# Adversarial Sample Detection via Channel Pruning

**Zuohui Chen** [* 1]  **RenXuan Wang** [* 1]  **Yao lu** [1]  **JingYang Xiang** [1]  **Qi Xuan** [1]

## Abstract

Adversarial attacks are the main security issue of deep neural networks. Detecting adversarial samples is an effective mechanism for defending against adversarial attacks. Previous works on detecting adversarial samples show superior in accuracy but consume too much memory and computing resources. In this paper, we propose an adversarial sample detection method based on pruned models. We find that pruned neural network models are sensitive to adversarial samples, i.e., the pruned models tend to output labels different from the original model when given adversarial samples. Moreover, the channel pruned model has an extremely small model size and actual computational cost. Experiments on CIFAR10 and SVHN show that the FLOPs and size of our generated model are only 24.46% and 4.86% of the original model. It outperforms the SOTA multi-model based detection method (87.47% and 63.00%) by 5.29% and 30.92% on CIFAR10 and SVHN, respectively, with significantly fewer models used.

## 1. Introduction

Though Deep Neural Networks (DNN) have achieved great success in various applications, e.g., computer vision (Pham et al., 2020), natural language processing (Brown et al., 2020), and speech recognition (Pan et al., 2020), the existence of adversarial samples undermines their use in safety critical areas and raises public concern. The Machine Learning (ML) community has proposed many approaches to improve DNN robustness against adversarial samples, including data augmentation (Tian et al., 2018; Kurakin et al., 2016), adversarial training (Yu et al., 2018; Li et al., 2020b), and robust optimization (Deng et al., 2020). These approaches can improve the robustness of the model to a cer-

tain extent, but ask for additional data and training, which cost intensive resources, especially for those large models.

The other optional defense strategy is detecting adversarial samples. The ML community has observed that adversarial samples are different from benign samples in multiple aspects, including data distribution (Li et al., 2020a; Chen et al., 2020), decision boundary (Yin et al., 2019), and neuron activating path (Ma & Liu, 2019). The model developer can distinguish adversarial samples by these characteristics and stop them from attacking the model. The software engineering community proposes the concept of DNN testing that aims to detect bugs in the DNN model (Tian et al., 2018; Ma et al., 2018; Wang et al., 2020). Adversarial samples are a kind of bug hidden in the DNN model. One of these testing methods for detecting adversarial samples is mutation testing (Wang et al., 2019). It generates multiple models by randomly shuffling neuron weights or change the activation state of neurons. They find that the generated models are sensitive to adversarial samples, which means the outputs of generated models are different from the original model. For an unknown sample, through the label changes of generated models, we can distinguish whether it is adversarial.

We argue that distinguishing adversarial samples using multiple models is a more reliable strategy for adversarial sample detection. (Huang et al., 2020) shows that many defense methods, including improving model robustness or detecting adversarial samples, are vulnerable to certain attacks. In extreme cases, such as the attacker grabs both model details and defense strategy, these methods can be even circumvented (Carlini & Wagner, 2017). Most of the detection methods evaluated in (Carlini & Wagner, 2017) use indicators from a single model. Since the indicator comes from the victim model and the perturbation updated in the attack also uses the output of the model or its gradient information, it is possible to design an adaptive attack strategy to circumvent these indicators (Carlini & Wagner, 2017) by adding constraints in the attack objective. For the multiple-model method, the indicator comes from various models, it is hard to guarantee that the perturbation works on all the models.

However, mutation testing is barely practical because the detection requires running dozens of models with almost the same size as the original model. In this paper, we find that models with channels being randomly pruned are more

---

[*]Equal contribution [1]Institute of Cyberspace Security, Zhejiang University of Technology, Hangzhou 310023, China. Correspondence to: Qi Xuan <xuanqi@zjut.edu.cn>.

*Accepted by the ICML 2021 workshop on A Blessing in Disguise: The Prospects and Perils of Adversarial Machine Learning.* Copyright 2021 by the author(s).

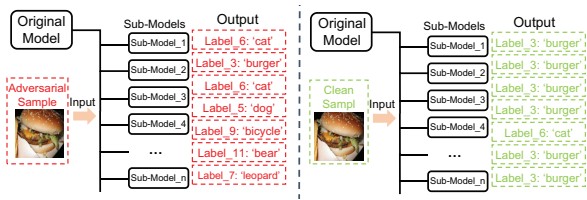

*Figure 1.* The outputs of benign sample and adversarial sample.

sensitive to adversarial samples. Moreover, pruning also reduces the model size, making the detection feasible in practice. Through experiments on CIFAR10 and SVHN, we proved the effectiveness of our method. It can detect 92.76% and 93.92% adversarial samples on CIFAR10 and SVHN, with 29.24 and 30.68 pruned models that are only 24.46% and 4.86% the size of the original models, respectively.

## 2. Related Works

As far as we know, Model Mutation Testing (MMT) (Wang et al., 2019) is the only approach to detect adversarial samples using multiple completely different models (not part of the original model). The other multi-model detection methods use models intercepted from the original model (Wang et al., 2020). The authors propose to build sub-models with the parameters and structure inherited from the original model and use them to detect adversarial samples. The number of sub-models can be as many as the number of intermediate layers of the original model. They argue that a normal sample should be predicted with increasing confidence, which reflects on the output of sub-models. Their method needs to retrain an output layer based on the inherited layers, but the inherited layers are frozen during training. In MMT (Wang et al., 2019), the authors propose to generate mutated models through four operators, namely weights fuzzing, weights shuffling, neuron switch, and neuron activation inverse. The operators will not cause a significant decrease in accuracy, but the generated models are sensitive to adversarial samples. Their method usually takes dozens of models to complete one detection.

## 3. Method

We detect adversarial samples by the outputs of pruned models. As shown in Figure 1, a benign sample with the label burger is still burger in most outputs of additional models, but an adversarial sample makes these models output various labels, e.g., cat, bicycle, bear. Because pruning and training rebuild the decision boundary of the pruned model, making it different from the original model. The diversity of outputs can be measured with Label Change Rate (LCR) and used to identify adversarial samples, which is defined as

$$\varsigma = \frac{\sum_{s \in S} E(f(x), s(x))}{|S|}, \tag{1}$$

where $x$ is the input, $f(x)$ is the original model output, $s(x)$ is the pruned-model output, $|S|$ is the size of used pruned models, $C$ is the number of classes, and $E(\cdot)$ is defined as

$$E(x, y) = \begin{cases} 0 \text{ if } x = y, \\ 1 \text{ otherwise.} \end{cases} \tag{2}$$

Instead of using the original model structure and parameters, we exploit random channel pruning to produce sub-models. Compared with the mutated models in Wang et al.'s work (Wang et al., 2019), the pruned model has a smaller size and is more sensitive to adversarial samples, which means our method requires fewer models and is also faster. Channel pruning usually evaluates the importance of different channels of a DNN layer and removes all the input and output connections of the unimportant channels (Gao et al., 2018; He et al., 2017; Zhuang et al., 2018). The advantages of channel pruning include reduction of actual parameters and increase of inference speed. Model pruning aims to find the smallest model with the least accuracy loss, while our work focuses on generating a model set for detection. Thus we use random channel pruning to find a set of small models with accuracy close to the original model.

In order to reduce the computational cost while ensuring the pruned models' accuracy and diversity, we set a fixed overall pruning rate for each model and assign a random number of channels that need to be pruned in each layer. Specifically, every several layers with the same number of channels are divided into a group. A group has an overall pruning rate (e.g., 50%) and layers in a group will be assigned with two random pruning rates (e.g., 30% and 20%), while the sum of which equals the overall pruning rate.

A straightforward way to calculate LCR is using the fixed-size sampling test, i.e., adopting a fixed number of models and counting the outputs that are different from the original model. To reduce the computational cost, we use the Sequential Probability Ratio Testing (Wald, 2004) (SPRT) to detect adversarial samples dynamically. The mutual exclusive hypothesis in the testing is

$$\begin{aligned} H_0 &: pr(x) \geq \varsigma_h, \\ H_1 &: pr(x) \leq \varsigma_h, \end{aligned} \tag{3}$$

where $\varsigma_h$ is a threshold determined by the LCR of benign samples (calculated by Eq. (1)). SPRT runs the pruned model successively and calculate the probability ratio $pr$ by

$$pr = \frac{p_1^z (1 - p_1)^{n-z}}{p_0^z (1 - p_0)^{n-z}}, \tag{4}$$

Table 1. Channel Pruning Rate of Different Groups.

| Number of Channels | 64 | 128 | 256 | 512 |
|---|---|---|---|---|
| Pruning Rate (%) | 50 | 60 | 70 | 80 |

Table 2. Average Model Size of MMT and Ours (MB).

|  | Original | Mutated | Pruned |
|---|---|---|---|
| ResNet18 | 85.35 | 85.35 | 20.88 |
| VGGNet16 | 112.45 | 112.45 | 5.47 |

Table 3. FLOPs of Generated Models.

|  | Original | Mutated | Pruned |
|---|---|---|---|
| ResNet18 | 140.60M | 140.60M | 51.82M |
| VGGNet16 | 314.03M | 314.03M | 32.42M |

where $z$ is the number of models that output different labels, $n$ is the total number of used models, $p_0 = \varsigma_h + \sigma$, and $p_1 = \varsigma_h - \sigma$. We set a relax scale $\sigma$, which means when the LCR falls in the region $(\varsigma_h - \sigma, \varsigma_h + \sigma)$, neither hypothesis can be denied and the test continues. The accept LCR and deny LCR are defined as follows

$$
\begin{aligned}
\varsigma_a &= ln\frac{\beta}{1-\alpha}, \\
\varsigma_d &= ln\frac{1-\beta}{\alpha},
\end{aligned}
\tag{5}
$$

where $\alpha$ and $\beta$ denote the probability of false positive and false negative, respectively. The test stops when one of the hypothesis is accepted. The input is considered as a benign sample if $pr \geq \varsigma_a$, while it is adversarial otherwise.

# 4. Experiments

## 4.1. Dataset and Models

We evaluate our approach on CIFAR10 and SVHN. The former contains 50,000 images for training and 10,000 images for testing, while the latter are 73,257 and 26,032, respectively. The image size of both two datasets is $32 \times 32 \times 3$. We adopt ResNet18 and VGGNet16 for CIFAR10 and SVHN, and their accuracy is 93.03% and 95.63%, respectively.

## 4.2. Random Channel Pruning

We set different pruning rates according to the number of channels of the layer in the group. The details are shown in Table 1, e.g., the layers in a group have 64 channels, the group pruning rate is set to 50%. Overall, 65% of the channels are pruned for ResNet18 and 70% for VGGNet16. Under this setting, the pruned model accuracy is above 90%, while the model size is relatively small. The average model size after pruning is listed in Table 2. We compare our approach with the MMT (Wang et al., 2019), which is a SOTA adversary detection algorithm. There are four mutation operators that can be used to generate mutated models, we choose the best performers for comparison, i.e., Neuron Activation Inversion (NAI). We use the best parameter setting, i.e., the mutation rate is 0.007. Both MMT and our method use SPRT to test the generated models' outputs. For a fair comparison, we set the maximum number of available models in SPRT to 100.

## 4.3. Adversarial Sample Generation

We use six typical adversarial attack methods, including 4 white-box and 2 black-box, each method generates 1,000 adversarial samples for detection. The parameters for each attack are summarized as follows:

1. FGSM: the scale of perturbation is 0.03;

2. JSMA: the maximum distortion is 12%;

3. CW: adopt L2 attack, the scale coefficient is 0.6 and the iteration number is 1000;

4. Deepfool (DF): the maximum number of iterations is 50 and the termination criterion is 0.02;

5. One Pixel Attack (OP): the number of pixels for modification is 3 (in order to ensure that enough successful samples are generated) and the differential algorithm runs with a population size of 400 and a max iteration count of 100;

6. Local Search Attack (LS): the pixel complexity is 1, the perturbation value is 1.5, the half side length of the neighborhood square is 5, the number of pixels perturbed at each round is 5 and the threshold for k-misclassification is 1.

In addition to the above seven attacks, we also regard the normal samples which Wrongly Labeled (WL) by the original model as adversarial samples.

## 4.4. Metrics

a) AUROC: Our approach takes the LCR of normal samples as the threshold. In order to verify whether the feature is suitable for distinguishing adversarial sample from the normal sample, we calculate the area under the ROC curve to determine whether LCR is an appropriate feature (the closer the AUROC is to 1, the better the feature is).

b) Detection accuracy and the number of models used: in addition to detection accuracy, we also evaluate the number of models required for detection. The higher the accuracy, the fewer models needed, the better the method.

*Table 4.* AUROC of Different Methods and Attacks.

| Dataset | Attack | MMT (NAI) | Ours |
|---------|--------|-----------|------|
| CIFAR10 | FGSM | 0.9214 | **0.9849** |
|  | JSMA | 0.9788 | **0.9921** |
|  | CW | 0.9170 | **0.9761** |
|  | DF | 0.9728 | **0.9933** |
|  | OP | **0.9977** | 0.9875 |
|  | LS | 0.9399 | **0.9569** |
|  | WL | 0.9238 | **0.9341** |
| SVHN | FGSM | 0.8761 | **0.9760** |
|  | JSMA | 0.9625 | **0.9893** |
|  | CW | 0.8922 | **0.9923** |
|  | DF | 0.9659 | **0.9973** |
|  | OP | 0.9497 | **0.9904** |
|  | LS | 0.9330 | **0.9862** |
|  | WL | 0.8902 | **0.9394** |

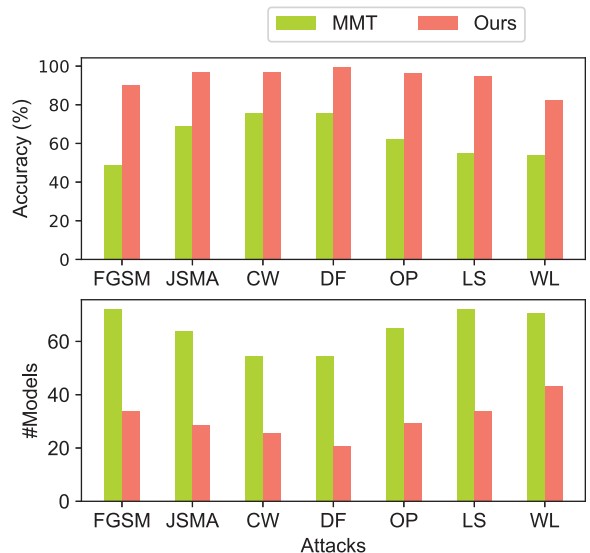

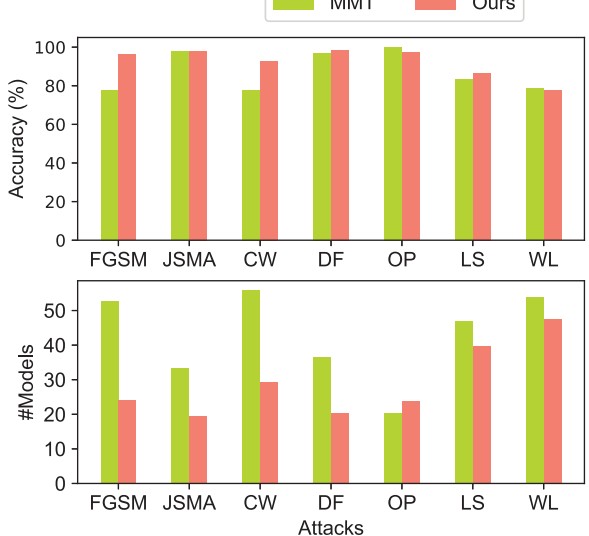

*Figure 2.* The detection accuracy (Accuracy) and the number of used models (#Models) on CIFAR10.

*Figure 3.* The detection accuracy (Accuracy) and the number of used models (#Models) on SVHN.

### 4.5. Results

As shown in Table 2, our random pruning strategy greatly reduced the model size from $85.35\,\text{MB}$ and $112.45\,\text{MB}$ to $20.88\,\text{MB}$ and $5.47\,\text{MB}$ on CIFAR10 and SVHN, respectively. The FLoating point OPerations (FLOPs) of generated models are listed in Table 3. The pruning reduces the number of FLOPs from 140.60M and 314.03M to 51.82M and 32.42M on CIFAR10 and SVHN respectively. Note that the mutation operator does not change the model size nor the FLOPs, thus models generated by MMT are the same size as the original model.

AUROC scores are summarized in Table 4. With the best results marked in bold, our approach outperforms MMT with average AUROC 0.9750 and 0.9816 on CIFAR10 and SVHN respectively, while MMT is 0.9502 and 0.9242. It shows that our pruned models are better in distinguishing

normal samples and adversarial samples.

Figure 2 and Figure 3 show the adversarial sample detection accuracy and the number of used models. On average, our approach used only 29.24 and 30.68 models for all attacks, while MMT requires 42.81 and 64.66 models for CIFAR10 and SVHN, respectively. In addition to the reduction in model numbers, our pruned model also has advantages in terms of memory footprint and inference speed. The average detection accuracy on CIFAR10 and SVHN are 92.76% and 93.92%, exceeding MMT (87.47% and 63.00%) 5.29% and 30.92%, which shows the superior of our approach.

## 5. Conclusion

In this paper, we propose an adversarial sample detection algorithm based on random channel pruning models. Compared with mutation operators, channel pruning greatly reduces the actual model size and improves the model sensitivity to the adversarial samples. We use SPRT to test the pruned models outputs and detect adversarial samples through the label changing rate. Experimental results show that our method outperforms MMT in AUROC, the number of used models, model size, and detection accuracy. The average AUROC of our method outperforms MMT (0.9502 and 0.9242) 0.0248 and 0.0574 on CIFAR10 and SVHN, respectively. Using only 29.24 and 30.68 models, the average detection accuracy of our method on CIFAR10 and SVHN are 92.76% and 93.92%, while MMT needs 42.81 and 64.66 models with accuracy 74.74% and 44.79%.

## Acknowledgements

This work was supported in part by the National Natural Science Foundation of China under Grant 61973273, and by the Zhejiang Provincial Natural Science Foundation of China under Grant LR19F030001.

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
