# OpenReview forum: "Adversarial Sample Detection via Channel Pruning"
_ICML.cc/2021/Workshop/AML — ICML 2021 Workshop AML Poster_

### Official Review · Reviewer_skeC · 2021-06-20
**More experiments on larger datasets make the paper more convincing**

**Rating:** Accept
**Confidence:** 3

**Review:**

Through pruning, this paper changes the decision boundary of models and makes the defense model more sensitive to adversarial images, and adversarial images can be detected by model ensemble. At the same time, pruning significantly reduces the size of the model, compared with other methods, it saves computation cost and improves detection performance.
I have a question:
The experimental part is only tested on relatively small datasets, which proves that the model can have a relatively large compression ratio. Can this ratio be replicated in large datasets? Or you can try the experiment on a dataset like tine-imageNet.

---

### Decision · Program_Chairs · 2021-06-21

**Decision:**

Accept (Poster)

**Comment:**

This paper proposed to detect adversarial examples by pruning. The authors can further address the reviewer's comments.